# Improved designs for pET expression plasmids increase protein production yield in *Escherichia coli*

Patrick J. Shilling [1✉], Kiavash Mirzadeh[1,2], Alister J. Cumming[1], Magnus Widesheim[1], Zoe Köck[1,3] & Daniel O. Daley[1✉]

The pET series of expression plasmids are widely used for recombinant protein production in *Escherichia coli*. The genetic modules controlling transcription and translation in these plasmids were first described in the 1980s and have not changed since. Herein we report design flaws in these genetic modules. We present improved designs and demonstrate that, when incorporated into pET28a, they support increases in protein production. The improved designs are applicable to most of the 103 vectors in the pET series and can be easily implemented.

[1] Department of Biochemistry and Biophysics, Stockholm University, Stockholm, Sweden. [2] Present address: Xbrane Biopharma, Solna, Sweden. [3] Present address: Goethe Universität, Frankfurt am Main, Germany. ✉email: patrick.shilling@dbb.su.se; ddaley@dbb.su.se

Studier and co-workers[1,2] described the first pET expression plasmid more than thirty years ago. They integrated the strong φ10 promoter for the T7 RNA polymerase (T7 promoter) and the Tφ transcription terminator (T7 terminator) into the pBR322 backbone and established the pET nomenclature (plasmid for expression by T7 RNA polymerase). Novagen and Invitrogen subsequently expanded the series to 103 unique expression plasmids. These expression plasmids support high levels of transcription in strains of *Escherichia coli* that contain a lysogenised *DE3* phage fragment encoding the T7 RNA polymerase and they have become a workhorse for the scientific community[3,4]. To date, they have been described in >220,000 published research studies (>12,000 per year; Supplementary Fig. 1).

pET28a is the most popular expression plasmid on the market (described in >40,000 published articles). It contains the T7 promoter and an adjacent *lac* operator sequence that is included to suppress uninduced expression[5]. Translation initiation is mediated by a Shine–Dalgarno (SD) sequence originating from the major capsid protein of T7 (*gene 10* protein). In a typical experiment, the coding sequence to be expressed is cloned downstream of, and in frame with, the coding sequence for a poly-histidine purification tag (His₆) and a thrombin protease recognition site (TPS) so that the recombinant protein produced can be easily purified using standardised protocols. The salient features of pET28a are presented in Fig. 1a.

In this study, we have identified design flaws in the pET series of expression plasmids, which limit protein production yields. We noted that (1) the T7 promoter consensus sequence was truncated in the T7*lac* promoter, and (2) that the translation initiation region (TIR) may have originally been formed by ad hoc genetic fusion. We describe solutions that rectify these design flaws, and demonstrate that when incorporated into pET28a, they increase protein production for three different proteins. The study therefore describes an easily implementable strategy for increasing recombinant protein yields using pET expression plasmids.

## Results

**The T7*lac* promoter lacks the complete T7 consensus sequence.** The first design flaw in pET28a is in the T7 promoter. The nucleotide sequence is derived from the consensus φ10 promoter in the T7 phage, which is 23 nucleotides long and sits −17 to +6 relative to the messenger RNA (mRNA) start site (Fig. 1b)[6]. We noted that pET28a only contains the −17 to +2 region, as four nucleotides were removed when the *lac* operator sequence was originally introduced in the early generation pET plasmids (designated T7*lac*)[5]. At the time it was stated that the *lac* operator has little effect on induced protein expression levels. Subsequent work suggested that divergence from the consensus T7 promoter (designated T7p^CONS) sequence decreases productive transcription initiation[7]. To determine if the +3 to +6 nucleotides are important in the context of pET28a, we compared the expression levels of the superfolder green fluorescent protein (His₆-TPS-sfGFP, hereafter referred to as sfGFP) from the commercially available pET28a and a variant where we had engineered in the T7p^CONS. We noted a three-fold increase in production yields of sfGFP in BL21(DE3) pLysS when T7p^CONS was used (Fig. 1c). Similar results were observed when using alternative strains such as C41 and C43[8] (Supplementary Fig. 2a, b) and when T7p^CONS was engineered into pET15b, which also includes the same T7*lac* promoter as pET28a (Supplementary Fig. 3a–c). The restoration of the T7 promoter to T7p^CONS did not change the sequence of the *lac* operator or its proximity from the coding sequence, and we still observed repression prior to induction (Supplementary Fig. 4). The experiment indicates that T7p^CONS is more efficient

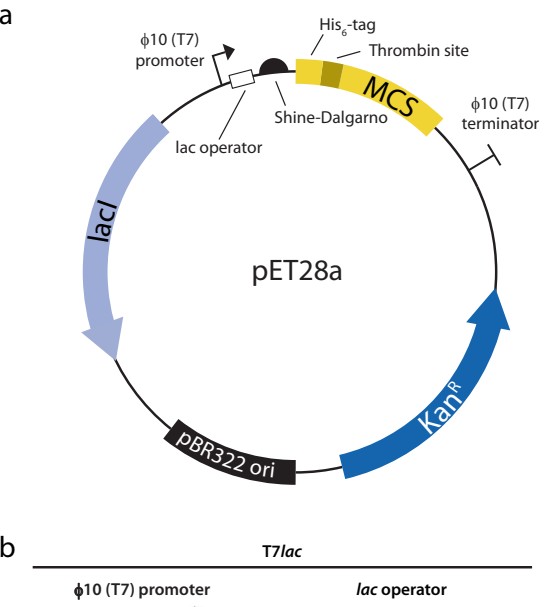

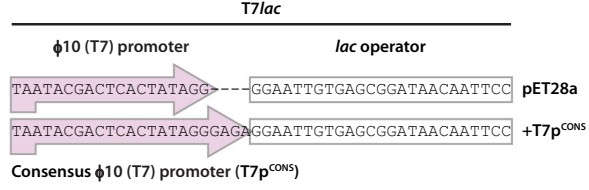

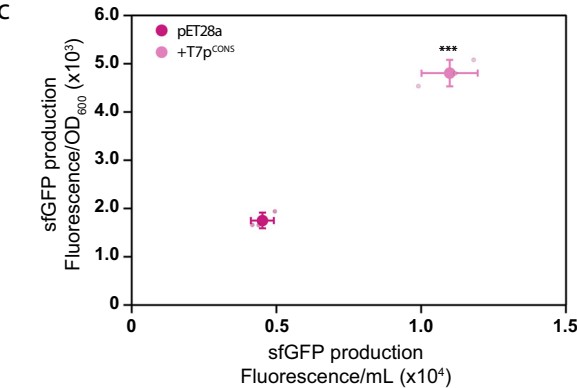

**Fig. 1 Salient features of pET28a, design flaws and improved designs. a** Genetic elements present in pET28a include the φ10 (T7) promoter and the *lac* operator, as well as the translation initiation region (TIR) encompassing the Shine–Dalgarno (SD) sequence, a spacer and the first five codons of the coding sequence. **b** The φ10 (T7) promoter in pET28a is a truncated variant of the consensus φ10 (T7) promoter (T7p^CONS). **c** Inclusion of the T7p^CONS results in a three-fold increase in sfGFP levels. Data are presented as mean ± s.d. (*n* = 3). A statistically significant difference of *p* < 0.001 relative to pET28a (two-tailed Student's *t* test) is denoted by ***.

than the truncated variant (−17 to +2) that is currently used in pET28a. The truncation of the T7 promoter is therefore a design flaw that reduces protein production. This design flaw is present in all pET expression plasmids containing T7*lac* (i.e. 88 of the 103 plasmids; Supplementary Fig. 1). In the remaining pET plasmids, the *lac* operator was not fused and the T7p^CONS is intact.

**Translation initiation is affected by ad hoc plasmid assembly.** The second design flaw in pET28a is in the TIR. The TIR is a stretch of ~30 nucleotides that is recognised by the 30S ribosomal subunit during translation initiation[9–11]. In a native *E. coli* mRNA, the TIR contains the SD sequence, a spacer that is between five and

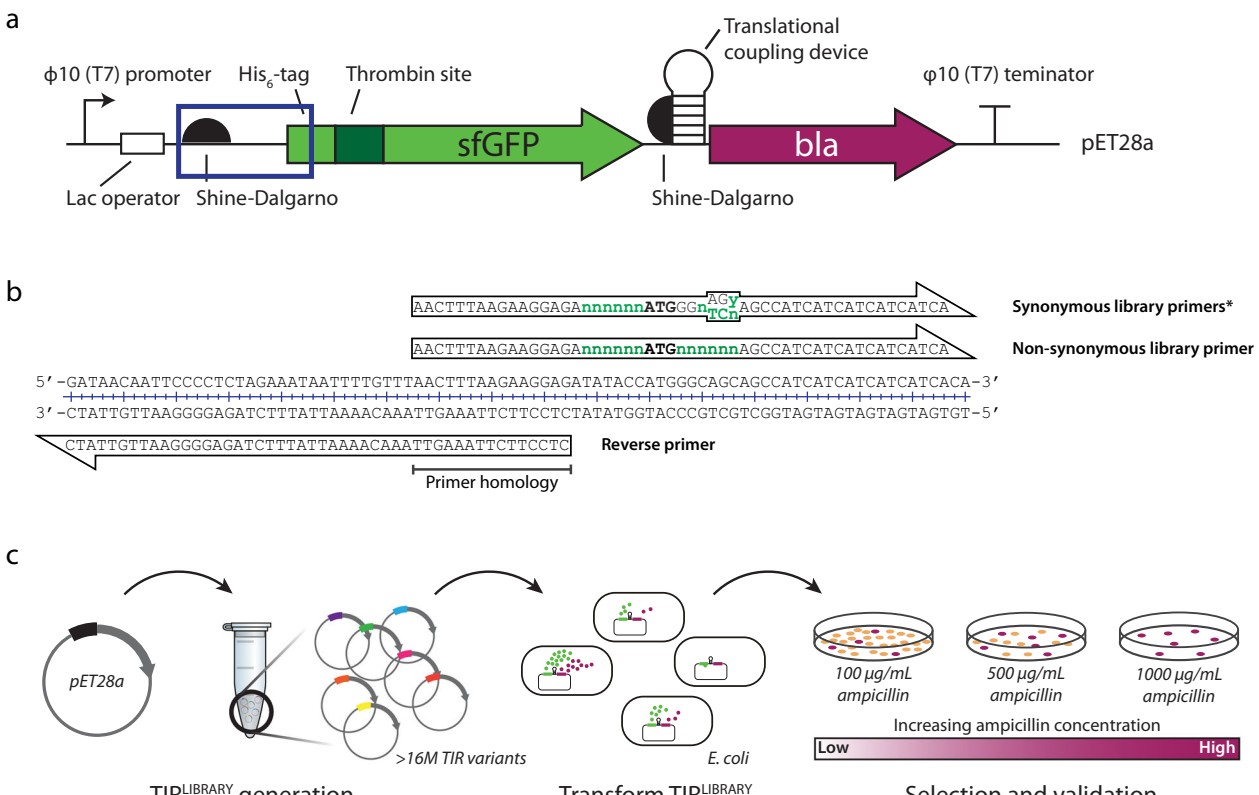

**Fig. 2 Method for synthetic evolution of the translation initiation region (TIR). a** Schematic of the pET28a-His$_6$-TPS-sfGFP-translational coupling device-β-lactamase expression cassette. The plasmid encodes for a His$_6$-tag, a thrombin protease site (TPS), sfGFP (green arrow), followed by a translational coupling device (hp; weak coupling 1) and the reporter, β-lactamase (bla; purple arrow). The boxed area represents the TIR for the expression cassette. **b** The pET28a-His$_6$-sfGFP-hp-bla plasmid was used as a template for the creation of TIR$^{LIBRARIES}$; degenerate primers used to create a TIR$^{LIBRARY}$ with synonymous codon changes and one with non-synonymous codon changes. **c** Following preparation of a TIR$^{LIBRARY}$ plasmids were transformed into *E. coli* BL21(*DE3*) pLysS. TIR$^{LIBRARY}$ variants that showed higher levels of protein expression are able to grow on plates at higher concentrations of ampicillin relative to the standard pET28a and were selected.

nine nucleotides in length[12,13], and the first five codons of the coding sequence (i.e. the first ribosomal footprint)[14,15]. Recent literature has indicated that native TIRs have co-evolved with the *E. coli* ribosomes and are less likely to be sequestered into local mRNA structures compared to the rest of the coding sequence[16,17]. The lack of mRNA structure is thought to promote the accessibility of the 30S subunit during translation initiation[18–21]. In pET28a the TIR is a composite of the SD sequence and a seven-nucleotide long spacer region from the major capsid protein of T7, and the first five codons of the plasmid encoded open reading frame (MGSSH). This region was constructed by Novagen and there is no publicly available literature describing its construction. We assume that it was assembled by ad hoc fusion of genetic modules rather than considering co-evolution with *E. coli* ribosomes. We therefore implemented a synthetic evolution approach to identify a TIR that was presumably more compatible with host cell ribosomes[22,23]. In the experiment, we used the standard pET28a as a template to generate two TIR libraries: one library covered >30,000 TIR variants and a second library that covered >16M possible TIR variants. We then tested the ability of TIR variants to support production of sfGFP by using a translational coupling device and β-lactamase[24] (Fig. 2). Initially, the TIR libraries were limited to synonymous codon changes at positions +2, +3 and we identified a TIR (TIR-1) that increased the production of sfGFP in BL21(*DE3*) pLysS by up to 13-fold (Fig. 3a, b). In a second experiment, we allowed all possible changes in codons +2, +3 and identified a TIR (TIR-2) that increased the production of sfGFP in BL21(*DE3*) pLysS by up

to 47-fold (Fig. 3a, b). It is unlikely that TIR-2 altered protein stability, as the original N-terminal amino acids (MG−) and the substituted amino acids (MQ−) are both considered stable according to the N-end rule[25]. Similar increases in production were observed when TIR-2 was used in the C41 and C43 strains (Supplementary Fig. 2c, d). In contrast to the synthetic evolution approach, attempts to identify optimal TIRs using bioinformatic algorithms were not successful (Supplementary Figs. 5 and 6). Taken together, these experiments indicate that the TIR in pET28a is not optimal for the production of sfGFP. The TIRs we identified are more effective in protein production experiments, and are directly applicable to pET14b, pET15b and pET28b–c, which possess the same TIR (Supplementary Fig. 7). Moreover, the synthetic evolution approach is applicable to all pET expression plasmids, as none, to our knowledge, have a TIR that has evolved with the host cell ribosomes.

**Implementation of design solutions in pET28a.** We created an improved version of the pET28a expression plasmid by engineering combinations of the T7p$^{CONS}$ together with either TIR-1 or TIR-2 (Fig. 4a). When combined we observed a 33- to 121-fold increase, respectively, in the synthesis of sfGFP compared to the standard version of pET28a (Fig. 4b). We also asked whether the improved version of the pET28a expression plasmid could increase protein production for alternative proteins. We tested the production yields for two human proteins, the putative cancer

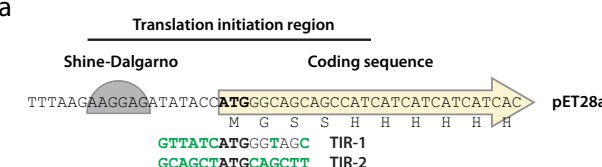

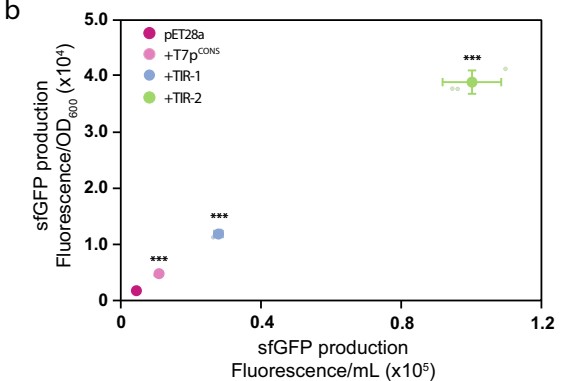

**Fig. 3 Synthetically evolved TIRs promote increased sfGFP production levels. a** Synthetic evolution of the pET28a-TIR resulted in two sequence variants (TIR-1 and TIR-2). Altered nucleotides for the TIR variants are shown in green text. **b** Inclusion of TIR-1 and TIR-2 resulted in up to a 13- and 47-fold increase in sfGFP production levels, respectively. Data are presented as mean ± s.d. ($n = 3$). A statistically significant difference of $p < 0.001$ relative to pET28a (two-tailed Student's $t$ test) is denoted by ***.

target MTH1[26,27] and the DNA glycosylase Neil3. We observed a greater than two-fold increase in protein production when we engineered T7p^CONS together with TIR-2 for both His6-TPS-MTH1 and His6-TPS-Neil3 (referred to as MTH1 and Neil3, respectively) (Fig. 4c, d). Varying results were observed however when either T7p^CONS or TIR-2 were used individually, revealing interesting context-specific expression profiles. The addition of T7p^CONS resulted in a marginal increase in overall production for MTH1 compared to the standard pET28a (Fig. 4c), whereas Neil3 showed a greater than three-fold increase (Fig. 4d). Engineering TIR-2 alone lead to two-fold increases for both MTH1 and Neil3 over the standard pET28a (Fig. 4c, d). Despite sharing identical promoters and TIRs, these results highlight the somewhat unpredictable nature of recombinant protein production.

Do increased production yields affect the solubility of the recombinant protein? We harvested cells expressing either sfGFP, MTH1 or Neil3 following induction from the commercially available pET28a expression plasmid, and then lysed and fractionated the cellular components into soluble and insoluble fractions. We observed that sfGFP and MTH1 were largely soluble, while Neil3 was largely insoluble (Fig. 5). We then repeated the experiment using the improved version of the pET28a expression plasmid, which contained T7p^CONS together with TIR-2. We again observed that sfGFP and MTH1 remained largely soluble, while Neil3 remained largely insoluble (Fig. 5). Consequently, the increased production yields obtained by incorporating more effective genetic modules into pET28a did not impact on overall protein solubility in the cell.

## Conclusions

Although the pET plasmid series work off the shelf, we have demonstrated that they contain design flaws in the genetic modules controlling transcription and translation initiation. We have identified improved designs that include (1) a restoration of the conserved T7 promoter (T7p^CONS) and (2) synthetically

evolved TIRs (TIR-1, -2). These improved designs work in combination to increase protein production yields in pET28a. They are easily incorporated and applicable across the majority of pET expression plasmids.

## Methods

**Google Scholar database searches.** Individual plasmid names from the pET (Novagen), pET (Invitrogen), pGEX (GE Healthcare), pQE (Qiagen) and pBAD (Invitrogen) plasmid series were queried in Google Scholar to determine the number of times each was used in a publication. For pET plasmids with multiple letter suffixes, the suffix would be included; for instance, pET28a, pET28b, pET28c and so on. The average number of times the pET series as a whole was used in publications per year was measured using a 5-year time span from 2014 to 2018. Searches included both publication in scientific journals and patents.

**Molecular cloning.** All polymerase chain reactions (PCR) were carried out with the Q5-polymerase (New England Biolabs, USA). Oligonucleotide synthesis and DNA sequencing was performed by Eurofins Scientific (Eurofins genomics, Germany). Oligonucleotides used for this study are found in Supplementary Table 1. Cloning of the coding sequence for sfGFP into pET28a was performed by standard restriction enzyme digestion and T4 ligation (New England Biolabs, USA) using the NdeI and XhoI sites, preserving the 5′ His6 and thrombin protease recognition coding sequence. The coding sequence for sfGFP was PCR amplified, incorporating the 5′ NdeI and 3′ XhoI restriction recognition sites as primer extensions. Generation of the pET28a-His6-TPS-sfGFP-translational coupling device (hp)-β-lactamase (bla) plasmid was carried out using the Gibson cloning method[28]. In brief, PCR products were generated for the pET28a-His6-TPS-sfGFP (described above) and β-lactamase, sourced from the pETDUET-1 template (Novagen, Germany). A translational coupling device (weak coupling 1)[24] was incorporated during the Gibson cloning step, via a 30-bp complementary overlapped region between the 3′ end of the pET28a-His6-TPS-sfGFP PCR product, and the 5′ end of the β-lactamase PCR product. All coding sequences are shown in Supplementary Figs 8–11. Human Neil3 (kindly provided by Pål Stenmark) was inserted into pET28a between the NdeI and XhoI restriction sites via Gibson assembly. All primer sequences are shown in Supplementary Table 1.

**Mutagenesis of pET28a φ10 promoter.** Mutagenesis of the φ10 promoter was carried out using the method of Liu and Naismith[29]. Briefly, the region encompassing the φ10-promoter initiator region (+2 to +6, GAGA) was incorporated into the 13 bp overlap of both the forward and reverse primer. The primers had sufficient complementarity to the template such that complementation to the template was favoured over primer dimer formation. Primer sequences are shown in Supplementary Table 1.

**sfGFP fluorescence assays.** Fluorescence assays were carried out as described[30] with minor modifications. Clones were transformed into chemically competent BL21(DE3) pLysS, C41 or C43. Three biological replicates were grown overnight at 37 °C with shaking at 180 RPM in 1 mL Luria–Bertani (LB) plus antibiotics in 96-well 2 mL deep-well culture plates. Overnight cultures were used to inoculate 5 mL LB plus antibiotics in a 24-well growth plate and incubated at 37 °C with shaking at 180 RPM until an OD600 of 0.5 was reached. Expression was induced by the addition of 1 mM isopropyl-β-D thiogalactopyranoside (IPTG) and cultures were incubated for 2 h at 37 °C with shaking at 180 RPM. The OD600 was measured, followed by collection of 1 mL of culture by centrifugation at 3220 × g for 15 min. The media were removed and pelleted cells were resuspended in 200 μL buffer (50 mM Tris-HCl, pH 8.0, 200 mM NaCl, 15 mM EDTA), and transferred to a 96-well optical bottom black-wall plate (Thermo Scientific). Following a 2-h incubation at room temperature, which enabled sfGFP to mature, fluorescence was read in a Spectramax Gemini (Molecular Devices) at an excitation and emission wavelength of 485 and 510 nm, respectively. Calculations of sfGFP yield per litre were determined based on a calibration curve, using purified sfGFP of known concentration. Quantification data are available in Supplementary Data 1 file (excel format).

**Generation of TIR libraries.** TIR libraries (TIR^LIBRARIES) were generated by amplifying either the pET28a-sfGFP-hp-bla expression plasmid by PCR, using overlapping primers as previously described[22,31]. For each library, the forward primer incorporated six degenerate nucleotides before the initiating start codon, and either synonymous or non-synonymous codon changes for the +2 and +3 codons (Gly, Ser). The reverse primer overlapped with the forward primer by 14 nucleotides, thus allowing circularisation of the PCR product by homologous recombination in the E. coli strain MC1061. The PCR was carried out using a programme that consisted of a denaturation step at 94 °C for 5 min, followed by 30 cycles of 95 °C for 45 s, 40–70 °C for 45 s (using a gradient thermocycler), 72 °C for 4 min and a final elongation step of 72 °C for 5 min. PCR products that were successfully amplified at the lowest annealing temperature with no contaminating non-specific PCR fragments were used for subsequent steps. Twenty-five microlitres of the PCR reaction was treated with DpnI, followed by transformation into 200 μL chemically competent E. coli MC1061 using standard protocols that

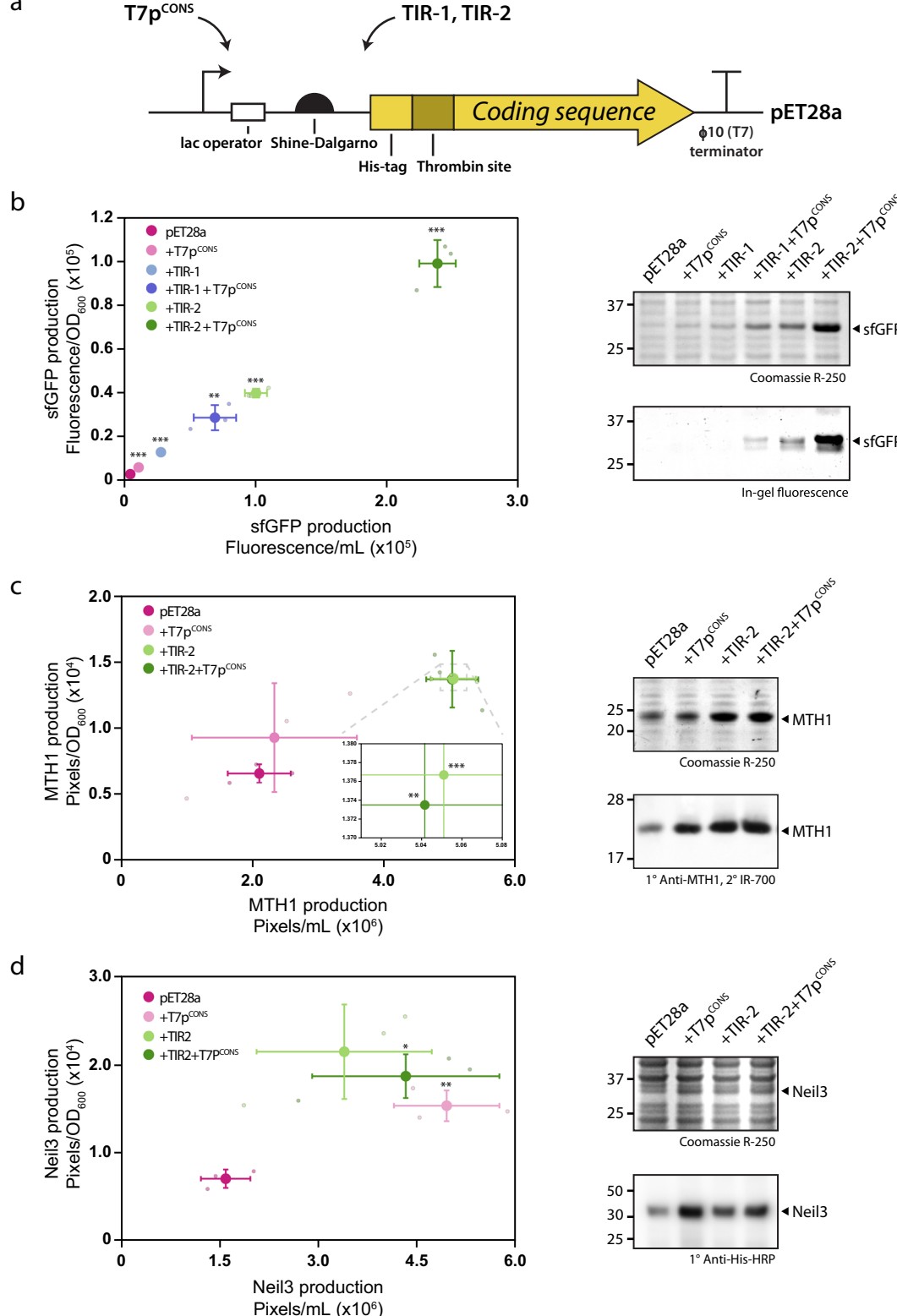

**Fig. 4 Combining the T7p$^{CONS}$ and TIR variants increases production efficiency. a** Schematic of the pET28a transcription and translation initiation region. In a standard production experiment, the coding sequence for the protein of interest is inserted in frame with the His$_6$-tag and thrombin protease site (His$_6$-TPS). **b** Engineering the T7p$^{CONS}$ in combination with either TIR-1 or TIR-2 resulted in an improvement in protein production. For sfGFP a maximal 121-fold increase was observable relative to the standard pET28a (left panel). sfGFP is the major protein constituent when T7p$^{CONS}$ and TIR-2 are combined (right panel). **c**, **d**. MTH1 and Neil3 production is enhanced by the addition of the combined TIR-2 and T7p$^{CONS}$. Data are presented as mean ± s.d. ($n = 3$). A statistically significant difference of $p < 0.05$, $p < 0.01$ or $p < 0.001$ relative to $p$ET28a (two-tailed Student's $t$ test) is denoted by *, ** and ***, respectively.

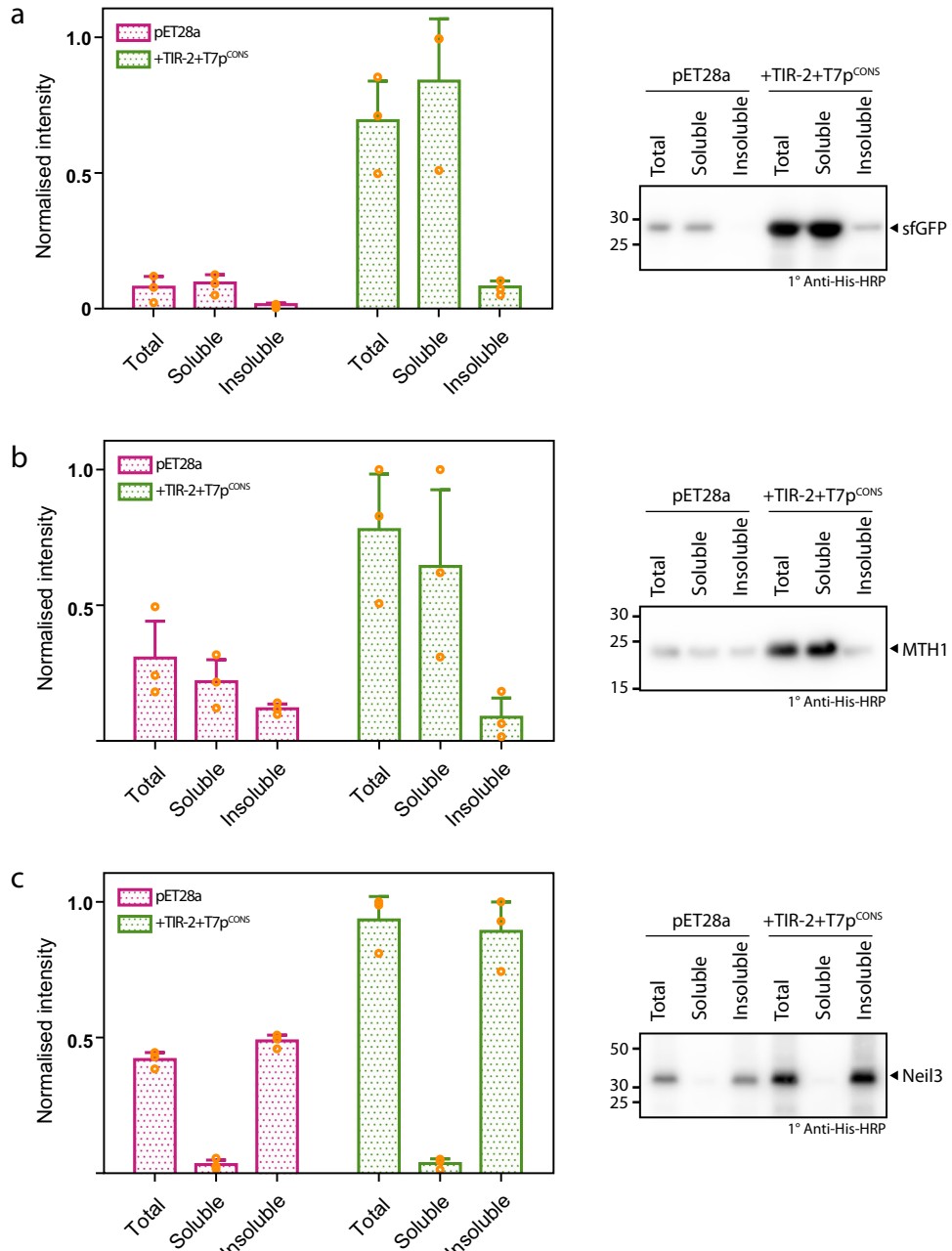

**Fig. 5 Cellular fractionation following expression from standard pET28a plasmid versus TIR-2 + T7p^CONS plasmid.** Western blot quantification of fractionations of BL21(*DE3*) *pLysS* (total, soluble and insoluble) following expression of **a** sfGFP, **b** MTH1 or **c** Neil3 from either the standard pET28a plasmid (pink bars) or pET28a-TIR-2+T7p^CONS plasmid (green bars). In all examples, application of the combined TIR-2+T7p^CONS resulted in enhanced levels of target proteins. Furthermore, use of TIR-2+T7p^CONS did not alter the solubility profile for each target. Representative Western blots are shown on the right panels with each protein marked. Data are presented as mean ± s.d. ($n = 3$).

included a 2-min heat shock at 42 °C and a 60-min recovery at 37 °C. The transformation was transferred into 4 × 5 mL LB media containing 50 µg/mL kanamycin contained in 4 × 50 mL conical tubes, and incubated overnight at 37 °C with shaking at 180 RPM. Isolation of the TIR^LIBRARIES was carried out using four E.Z. N.A DNA mini kit purification columns (Omega Bio-tek, USA) according to the manufacturer's instruction, followed by pooling of the eluates. All primer sequences for the generation of TIR^LIBRARIES are shown in Supplementary Table 1.

**Screening of TIR libraries**. TIR^LIBRARIES were screened by transforming chemically competent BL21(*DE3*) *pLysS* using standard protocols and comparing colony formation on plates containing increasing amounts of ampicillin. Specifically, 500 ng of the TIR^LIBRARY was transformed into 50 µL of chemically competent BL21(*DE3*) *pLysS* using standard protocols. Following recovery, the whole

transformation mixture was seeded into 3 mL of LB contained in a 15 mL conical tube, followed by incubation at 37 °C with shaking at 180 RPM for 16 h. The overnight culture was used to inoculate (1:50) 5 mL of fresh LB containing 50 µg/mL kanamycin and 34 µg/mL chloramphenicol. Cultures were grown until reaching an OD$_{600}$ of 0.3, whereupon expression of the coding sequence was induced by plating a volume of cells corresponding to 0.01 OD$_{600}$ units onto LB-agar plates (45 mm diameter) containing 0.25 mM IPTG, and increasing concentrations of ampicillin (100–4000 µg/mL). In order to ensure selection of optimal clones, based only on β-lactamase production, we selected clones only using ampicillin. Kanamycin and chloramphenicol, which are used to maintain pET28a and pLysS respectively, were omitted from the LB-agar plates. We reasoned that the use of kanamycin and chloramphenicol would apply too great a selection pressure on developing colonies. While not necessarily an important step, we have noted it so that readers can replicate our experiments. The plates were then incubated for ~120 h at 20 °C. A

TIR$^{LIBRARY}$ was deemed successful if colonies were capable of withstanding concentrations of ampicillin higher than colonies harbouring a plasmid with the standard TIR. For plates where TIR$^{LIBRARY}$ variants showed greater resistance to ampicillin compared to a standard TIR, ten colonies were selected for further analysis and sequencing (Eurofins MWG operon, Germany). Determination of sfGFP production was carried out in triplicate by the previously described fluorescence assay. Determination of MTH1 production levels was performed by Western blotting. For both protein targets, the best five clones exhibiting high levels of protein production were carried onto the next stage of evaluation. To exclude the possibility that mutations away from the TIR promoted higher expression levels, the best five selected expression variant TIR sequences were back-engineered into the original pET28a-His$_6$-TPS-sfGFP-hp-bla and pET28a-His$_6$-TPS-sfGFP expression vectors, and sfGFP assay or Western blot performed, respectively. The best TIR variant was then chosen for subsequent testing. The primer sequences for generating TIR-1 or TIR-2 (sfGFP) are shown in Supplementary Table 1.

**Protein expression and fractionation**. Overnight cultures (100 mL LB supplemented with 50 µg/mL kanamycin and 34 µg/mL chloramphenicol) were inoculated from freshly transformed BL21(DE3) pLysS expressing sfGFP, MTH1 or Neil3 in either the standard pET28a or pET28a-TIR-2+T7p$^{CONS}$ plasmid. Overnight cultures were grown at 37 °C with shaking at 200 RPM. The following morning, the overnight culture was used to inoculate 500 mL LB supplemented with 50 µg/mL kanamycin and 34 µg/mL chloramphenicol, to a starting OD = 0.05. Cultures were grown at 37 °C with shaking at 200 RPM until reaching an OD$_{600}$ = 0.5–0.7. Specifically, for sfGFP and Neil3, induction was induced immediately with 1 mM IPTG and allowed to incubate at 37 °C with shaking at 200 RPM for 2 h. For MTH1, the medium was cooled by incubation at 4 °C for 10 min, followed by induction with 1 mM IPTG and subsequent incubation at 16 °C with shaking at 200 RPM, overnight for 20 h. Following the allotted induction time, cells were harvested at 4000 × $g$ for 20 min at 4 °C. Bacterial pellets were resuspended in 50 mL standard buffer (20 mM Tris (pH 7.0), 150 mM NaCl). Cell pellet resuspensions were made homogeneous with a glass Dounce homogeniser. Resuspended cells were lysed by three passes at 10,000–15,000 PSI in an Avestin emulsiflex C3 high-pressure homogeniser (Avestin, Canada). Following collection of a total lysate sample, lysed cells were then centrifuged at 22,000 × $g$ for 1 h. The supernatant (soluble fraction) was collected by gentle aspiration and pellet fractions (insoluble fraction) resuspended in an equivalent volume of standard buffer. Samples were collected and run on a 12% sodium dodecyl sulfate-polyacrylamide gel electrophoresis (SDS-PAGE) gel followed by Western blotting. All constructs and conditions were carried out in triplicate.

**SDS-PAGE and Western blotting**. SDS-PAGE was carried out on a 4–12% Bis-tris Midi Protein Gel in an XCell4 SureLock Midi system (Invitrogen, USA) or 12% Bis-tris acrylamide using a Hoefer Mighty Small II Mini Vertical Electrophoresis System with a 1 mm thickness. The running buffer used was premixed with NuPAGE MES SDS (Invitrogen, USA). Samples consisted of whole-cell lysates or fractionated cells (total, soluble, insoluble), and were prepared first by incubation at 95 °C for 10 min, or 65 °C for sfGFP to preserve the folded fluorescent form used for in-gel fluorescence. For normalised consistent loading, the equivalent of 0.1 or 0.05 OD$_{600}$ units were applied to each well. Western blotting was carried out on PVDF or nitrocellulose membranes using either an XCell-II or an iBlot Module (Invitrogen, USA), respectively.

For PVDF membranes probed against the His$_6$ epitope: Gels transferred to PVDF were incubated in transfer buffer (Towbin buffer: 25 mM Tris, 192 mM glycine, 20% (v/v) methanol) for 15 min, followed by transfer to PVDF for 1 h at 30 V. Membranes were incubated in 5% skim milk powder (PanReac AppliChem) in TBS (50 mM Tris, pH 7.4, 200 mM NaCl) for 1 h at room temperature. Membranes were decorated with HisProbe-HRP conjugate (Thermo Scientific, USA) at 1:10,000. Membranes were developed with SuperSignal West Pico PLUS Chemiluminescent Substrate (Thermo Scientific, USA) on an Azure c600 Western Blot Imaging System (Azure Biosystems, USA). For nitrocellulose membranes probed against MTH1: Proteins were transferred to nitrocellulose for 7 min at 20 V using an iBlot Module. Membranes were incubated in 5% skim milk powder (Bio-Rad, USA) for 1 h at room temperature. Membranes were decorated with primary mouse anti-MTH1 antibody (MABC1040, Millipore) at 1:1000 dilution for 1 h at room temperature, followed by decoration with secondary donkey anti-mouse IR-680 antibody at 1:10,000 (LI-COR, USA). Nitrocellulose membranes were developed on an Odyssey Imaging System (LI-COR, USA). Western blots were analysed by Image J. Original uncropped gels and blots are presented in Supplementary Figs. 12–17. Quantification data are available in Supplementary Data 1 file (excel format).

**In silico TIR predictions**. Three prediction algorithms were used for the creation of in silico TIRs (RBS calculator[32], UTR designer[33] and RBS designer[34]). For RBS calculator, version 2.1 was tested. Presets included: pre-sequence position: −70 to −30; coding sequence from position 1 until +60; goal: maximise; organism: E. coli BL21(DE3) (ACCTCCTTA). UTR designer: constraints for 5′-UTR: 25 × N; coding sequence: position 1 until +60; desired protein level: maximal; organism: E. coli (ACCUCCUUA); with no optimisation of codon content. RBS designer: 5′-UTR: position −70 until −24; optimise upstream length: ten nucleotides (default); SD:

AAGGAA (default); optimise spacer length: seven nucleotides; coding sequence: position +1 until +60; organism: E. coli; target efficiency: 1.0. The best three variant sequences for each programme were utilised for the creation of in silico predicted TIRS in pET28a-sfGFP. Mutagenesis primers were designed and incorporated into pET28a-sfGFP using the previously described mutagenesis methodology. All primer sequences for generation of in silico predicted TIRS are shown in Supplementary Table 1.

**Statistics and reproducibility**. All experiments where statistical calculations were applied used three biological replicates. All data are presented as the mean ± the standard deviation. Statistical analysis was performed by the Student's $t$ test. $P$ values considered statistically significant are indicated by asterisk symbols within the figure legends.

**Reporting summary**. Further information on research design is available in the Nature Research Reporting Summary linked to this article.

## Data availability

All relevant data are available from the corresponding authors upon request. Source data underlying plots shown in figures are presented in Supplementary Data 1. Full blots are shown in Supplementary Information.

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

## Acknowledgements
We thank Pål Stenmark for the pET28a-His₆-TPS-MTH1 and pET28a-Neil3 constructs and Jan-Willem de Gier for the C41 and C43 strains. D.O.D. was supported by a grant from the Swedish Research Council and the Carl Trygger stiftelse. Open access funding provided by Stockholm University.

## Author contributions
P.J.S., K.M., M.W. and D.O.D. designed the study. P.J.S., K.M., A.J.M. and Z.K. carried out the experiments. P.J.S. and D.O.D. wrote the paper.

## Competing interests
The synthetic evolution process used in the study is patent protected (PCT/SE2015/051343; European Patent no. 3234146). The patent is the property of CloneOpt AB, of which K.M. and D.O.D. are shareholders. All the other authors declare no competing interests.
