## [Peer Review File · Communications Biology]

Reviewers' comments:

Reviewer #1 (Remarks to the Author):

The manuscript by Shilling et al. is entitled "Improved designs for pET expression plasmids". In this work, the authors exposed design flaws in pET plasmids which were created when the plasmids were constructed. The authors claim that these flaws are the cause for suboptimal protein expression. So, they corrected them using standard cloning procedures (in the case where the T7 promoter was "fixed") and a synthetic evolution approach based on a translational coupling device (in the case where the TIR was improved).

They show a remarkable increase in the expression of the target protein (sfGFP and later, MTH-1) when different genetic elements were used alone or in combination.

The work is attractive for researchers working in the broad field of heterologous protein production and in part it seems easy to implement (especially, the T7pCONS promoter). Experiments were performed using only two proteins and for one of them (MTH-1), a customized TIR was used. Protein content was measured in whole cell lysates, soluble protein content would have been more useful. No explanation was given as to why the expression of MTH-1 was customized. Also, the improved genetic elements are located upstream and inside the coding sequence so it is not clear if the modifications (namely, in the TIR) can be universally applied, as researchers may not want to alter the N-terminal end.

The manuscript is succinct and very well-written. I appreciate that the Methods section gives great detail so that experiments can be replicated. I hereby list some concerns that should be addressed by the authors before acceptance. They are listed in order of appearance.

1. "...in the pET series and are easily implemented...". Consider changing tense to "...can be easily..."
2. "To date they have been used in >220,000 published research studies (>12,000 per year; Supplementary Fig. 1)" Have the authors checked that the particular plasmid was in fact used in the study? If the plasmid was mentioned in the paper but not in fact used in experiments, would that count as a hit?
3. "The salient features of pET28a are presented in Fig. 1a." In the legend for Panel 1a, the abbreviation for TIR appears but it is defined later. If readers check the legend before reaching the fourth paragraph, they may not understand what TIR stands for. Consider introducing the concept sooner or defining it in the legend.
4. "...subsequent work suggested that divergence from the consensus T7 promoter sequence decreased productive transcription initiation⁷." The provided citation does not seem to support that claim (the paper describes the C41 and C43 strains).
5. "Similar results were observed using the pET15b plasmid (Supplementary Fig. 2) as well as when we used alternative strains such as C41 and C437 (Supplementary Fig. 3)" Please, point to the exact panel that illustrates the finding (in all instances of the manuscript).
6. "...spacer that is typically nine nucleotides in length..." The spacer used in this study is 7-bp long. Have the authors considered adding the missing 2-bp?
7. "To our knowledge, the pET28a TIR was assembled by ad hoc fusion of genetic modules..." Could that part be expanded describing the assembly of pET28 or cite the relevant paper?
8. "...the TIR in pET28a is not optimal for protein production." Up to this point, only sfGFP had been used. The conclusion strikes me as too universal considering that only one protein was tested. Please, rephrase.
9. "...a TIR (TIR-3) that was customized for MTH1..." Do the authors suggest that TIRs should be optimized in a case-by-case basis? Maybe a sentence should alert the reader that TIRs can be further optimized depending on the particular case.
10. "Generation of the pET28a-His6-TPS-sfGFP-hp-AmpR..." Does hp stand for something? Please, define.
11. "(weak coupling 1)" is called weak-1 in the legend of supp fig 5. Please, check consistency.

12. "Following a two-hour incubation at room temperature..." Why is this step necessary? Can't fluorescence be measured right away?
13. "...using purified sfGFP of known concentration." So, the authors are comparing in vivo fluorescence to fluorescence of purified sfGFP. Does sfGFP fluorescence behave the same under those two conditions?
14. "...incorporated six degenerate nucleotides..." The linker in pET28 is 7-bp long. Why was the first A left out? Also, the +4 codon was also left out.
15. "...competent E. coli MC1061. The transformation was..." Was there a recovery time in between those steps?
16. "Note that kanamycin and chloramphenicol were omitted from the plates at this stage." Why is this important?
17. "Samples consisted of whole cell lysates" Major issue here: we all aim to obtaining a lot of protein in the SOLUBLE fraction because a lot of protein in the whole lysate may just be insoluble. Consider performing at least one experiment showing the distribution of both GFP and MTH-1 in the soluble and insoluble fractions. I ask for both proteins as GFP is a well-behaved protein: it is probably soluble. MTH-1 is far more interesting in this case.
18. "SD: AAGGAA (default)" Is this option customizable? Because the SD in pET28a is AAGGAG.
19. Supp Fig 2, panel b: Please change pET28a to pET15b. Yes, they have the same sequence but that figure is representing pET15b.
20. Supp Fig 7, panel a: Major issue that should also be discussed in the main text. In the new TIRs, non-synonymous changes are introduced at the N-terminal end of the protein. Can this change its stability according to the N-end rule? What about the action of MetAPs? Can part of the results be a consequence of enhanced protein stability or diminished degradation instead of the phenomenon proposed by the authors (lack of mRNA structure and accessibility of the 30S subunit, etc.)?
21. Supp Fig 7. Panel b: why was protein/mL not measured? Fluorescence/OD may be higher using the optimized plasmids but if the culture grew poorly, then the significance of the improvement is reduced. The authors should produce a plot like the others, in which both yields (OD-normalized and per mL) were informed.

Reviewer #2 (Remarks to the Author):

The authors have identified in the pET plasmids, regions in the sequences controlling transcription and translation initiation that can be improved to optimise protein production. They describe a truncated T7 promoter region, which when corrected to contain the additional bases, improves expression of GFP by 3-fold. This was confirmed using two different pET vectors and three E. coli strains. The authors also looked at the Translation Initiation Region (TIR), suggesting that it was generated by fusion of genetic modules without the consideration of co-evolution with E. coli ribosomes. By altering this sequence randomly, they have generated TIRs that improve production of GFP. By combining the altered sequences for the T7 promoter and the TIRs, they found plasmids capable of increasing expression of GFP and a human cancer target, MTH1. The authors mention that these alterations to the pET vectors are applicable to 88 out of the 103 versions available.

This work is very interesting for the protein research community and could benefit many labs to improve production of their proteins in the E.coli expression system. The idea of changing regions in the plasmid to improve transcription and translation could also be applied to other vectors and protein expression systems.

Although the authors have shown that the method works for both a common control protein (GFP) and also a human protein (MTH1), the affect on a wider variety of proteins would be desirable.

In supplementary Figure 1, where the plasmids do not contain an asterix, does this mean that there is not a truncated T7 promoter and if so, have any of these plasmids been tested during the study?

Some minor corrections:

Online Methods:

sfGFP fluorescence assays:

line 5: incubated at 37

line 8: medium

Generation of TIR libraries:

line 12: medium

Screening of TIR libraries:

line 5: space between 37 degrees

line 14: plasmid

SDS-PAGE and Western blotting:

line 8: transferred

Reviewers' comments:

Reviewer #1 (Remarks to the Author):

The manuscript by Shilling et al. is entitled “Improved designs for pET expression plasmids”. In this work, the authors exposed design flaws in pET plasmids which were created when the plasmids were constructed. The authors claim that these flaws are the cause for suboptimal protein expression. So, they corrected them using standard cloning procedures (in the case where the T7 promoter was “fixed”) and a synthetic evolution approach based on a translational coupling device (in the case where the TIR was improved).

They show a remarkable increase in the expression of the target protein (sfGFP and later, MTH-1) when different genetic elements were used alone or in combination.

The work is attractive for researchers working in the broad field of heterologous protein production and in part it seems easy to implement (especially, the T7pCONS promoter). Experiments were performed using only two proteins and for one of them (MTH-1), a customized TIR was used. Protein content was measured in whole cell lysates, soluble protein content would have been more useful. No explanation was given as to why the expression of MTH-1 was customized. Also, the improved genetic elements are located upstream and inside the coding sequence so it is not clear if the modifications (namely, in the TIR) can be universally applied, as researchers may not want to alter the N-terminal end. The manuscript is succinct and very well-written. I appreciate that the Methods section gives great detail so that experiments can be replicated. I hereby list some concerns that should be addressed by the authors before acceptance. They are listed in order of appearance.

Author: First of all, we are grateful to the reviewer for providing feedback on our manuscript; It was extremely useful. We would like to report back that we have addressed the points raised in the sections below.

1. “...in the pET series and are easily implemented...”. Consider changing tense to “...can be easily...”

Author: Done

2. “To date they have been used in >220,000 published research studies (>12,000 per year; Supplementary Fig. 1)” Have the authors checked that the particular plasmid was in fact used in the study? If the plasmid was mentioned in the paper but not in fact used in experiments, would that count as a hit?

Author: Unfortunately we cannot say with certainty that the plasmid has been used in all publications, only that it has been cited. The text has therefore been modified to reflect this fact. It now reads “pET28a is the most popular expression plasmid on the market (described in >40,000 published articles).”

3. “The salient features of pET28a are presented in Fig. 1a.” In the legend for Panel 1a, the abbreviation for TIR appears but it is defined later. If readers check the legend before reaching the fourth paragraph, they may not understand what TIR stands for. Consider introducing the concept sooner or defining it in the legend.

Author: This has been corrected in the figure legend.

4. "...subsequent work suggested that divergence from the consensus T7 promoter sequence decreased productive transcription initiation⁷." The provided citation does not seem to support that claim (the paper describes the C41 and C43 strains).

Author: The reference has been corrected.

5. "Similar results were observed using the pET15b plasmid (Supplementary Fig. 2) as well as when we used alternative strains such as C41 and C437 (Supplementary Fig. 3)" Please, point to the exact panel that illustrates the finding (in all instances of the manuscript).

Author: This has been corrected.

6. "...spacer that is typically nine nucleotides in length..." The spacer used in this study is 7-bp long. Have the authors considered adding the missing 2-bp?

Author: The reviewer has raised an interesting point. Spacer lengths in *E. coli* range from 5-9 nucleotides in length. As the pET28a spacer is within this range (i.e. 7 nucleotides) we did not explore different lengths. But it is something that would be worth considering in future work.

7. "To our knowledge, the pET28a TIR was assembled by ad hoc fusion of genetic modules..." Could that part be expanded describing the assembly of pET28 or cite the relevant paper?

Author: Unfortunately there is not much more information to add, as this region was constructed by Novagen (a company). We have therefore adjusted the text to explain this fact. It now reads, "This region was constructed by Novagen and there is no publicly available literature describing its construction. We assume, that it was assembled by *ad hoc* fusion of genetic modules rather than considering co-evolution with *E. coli* ribosomes."

8. "...the TIR in pET28a is not optimal for protein production." Up to this point, only sfGFP had been used. The conclusion strikes me as too universal considering that only one protein was tested. Please, rephrase.

Author: We have corrected the text. It now reads, "Taken together, these experiments indicate that the TIR in pET28a is not optimal for production of sfGFP."

9. "...a TIR (TIR-3) that was customized for MTH1..." Do the authors suggest that TIRs should be optimized in a case-by-case basis? Maybe a sentence should alert the reader that TIRs can be further optimized depending on the particular case.

Author: We have now removed the experiments describing the customisation of a TIR (TIR-3) for MTH1, as they did not contribute to the overall message of the manuscript (and obviously caused some confusion).

10. "Generation of the pET28a-His6-TPS-sfGFP-hp-AmpR..." Does hp stand for something? Please, define.

Author: The text has been corrected. It now reads “Generation of the pET28a-His₆-TPS-sfGFP-translational coupling device-β-lactamase (bla) plasmid was carried out using the Gibson cloning method.”

11. “(weak coupling 1)” is called weak-1 in the legend of supp fig 5. Please, check consistency.

Author: This has been corrected in the figure legend (see Figure 2 in new version).

12. “Following a two-hour incubation at room temperature...” Why is this step necessary? Can't fluorescence be measured right away?

Author: In previous work, we have noted that the GFP fluorescence continues to rise after the cells have been harvested. This is presumably caused by the relatively slow maturation time of the chromophore in GFP. We therefore include a two hour incubation so that fluorescence levels have plateaued (or stabilised) and our assays are more reproducible. We have included a brief statement to this effect in the methods section.

13. “...using purified sfGFP of known concentration.” So, the authors are comparing in vivo fluorescence to fluorescence of purified sfGFP. Does sfGFP fluorescence behave the same under those two conditions?

Author: We do believe that purified sfGFP is a good approximation for the in vivo sfGFP. We have carried out a fluorescence curve to ensure that we are recording within the dynamic range of the instrument. Moreover we found that there was no dampening of fluorescence when cells were sufficiently diluted with the described buffer.

14. “...incorporated six degenerate nucleotides...” The linker in pET28 is 7-bp long. Why was the first A left out? Also, the +4 codon was also left out.

Author: The TIR randomisation process implemented in this study was one that we have developed previously (see references 23 and 24). Larger libraries, which include the first A and / or the 4th codon could also be considered.

15. “...competent E. coli MC1061. The transformation was...” Was there a recovery time in between those steps?

Author: We have now included this information in the Methods section. It reads “...using standard protocols that included a two minute heat shock at 42 °C and a 60 minute recovery at 37 °C.”

16. “Note that kanamycin and chloramphenicol were omitted from the plates at this stage.” Why is this important?

Author: We have included a statement that explains this situation. It reads “In order to ensure selection of optimal clones, based only on β-lactamase production, we selected clones only using ampicillin. Kanamycin and chloramphenicol, which are used to maintain pET28a and pLysS respectively, were omitted from the LB-agar plates. We reasoned that the use of kanamycin and chloramphenicol would apply too great a selection pressure on developing

colonies. While not necessarily an important step, we have noted it so that readers can replicate our experiments.”

17. “Samples consisted of whole cell lysates” Major issue here: we all aim to obtaining a lot of protein in the SOLUBLE fraction because a lot of protein in the whole lysate may just be insoluble. Consider performing at least one experiment showing the distribution of both GFP and MTH-1 in the soluble and insoluble fractions. I ask for both proteins as GFP is a well-behaved protein: it is probably soluble. MTH-1 is far more interesting in this case.

Author: We have fractionated *E. coli* cells expressing GFP and MTH-1 into soluble and insoluble fractions. These new data indicate that both proteins are predominantly in the soluble fraction. Thus the increased production yields obtained in the improved pET28a vectors did not impact on protein solubility. We have also tested another human protein, the DNA glycosylase Neil3. Neil3 was initially insoluble, and the increased production resulted in more insoluble protein. Consequently the increased production yields obtained by incorporating more effective genetic modules into pET28a did not impact on overall protein solubility in the cell. This new data has been incorporated into the revised manuscript (see text and new Figure 5).

18. “SD: AAGGAA (default)” Is this option customizable? Because the SD in pET28a is AAGGAG.

Author: This is an interesting point raised by the reviewer. We believe that this option was customisable in the software, but we chose to run all predictions in the ‘default settings’ so that performance of all programs could be fairly assessed.

19. Supp Fig 2, panel b: Please change pET28a to pET15b. Yes, they have the same sequence but that figure is representing pET15b.

Author: This figure has been corrected.

20. Supp Fig 7, panel a: Major issue that should also be discussed in the main text. In the new TIRs, non-synonymous changes are introduced at the N-terminal end of the protein. Can this change its stability according to the N-end rule? What about the action of MetAPs? Can part of the results be a consequence of enhanced protein stability or diminished degradation instead of the phenomenon proposed by the authors (lack of mRNA structure and accessibility of the 30S subunit, etc.)?

Author: This is a good point raised by the reviewer, so we have added a statement that clarifies how we interpret the data. It states “It is unlikely that TIR-2 increased protein stability, as the original N-terminal amino acids (MG-) and the substituted amino acids (MQ-) are all considered stable according to the N-end rule.”

21. Supp Fig 7. Panel b: why was protein/mL not measured? Fluorescence/OD may be higher using the optimized plasmids but if the culture grew poorly, then the significance of the improvement is reduced. The authors should produce a plot like the others, in which both yields (OD-normalized and per mL) were informed.

Author: We have adjusted the figure (see new Supplementary Figure 6).

Reviewer #2 (Remarks to the Author):

The authors have identified in the pET plasmids, regions in the sequences controlling transcription and translation initiation that can be improved to optimise protein production. They describe a truncated T7 promoter region, which when corrected to contain the additional bases, improves expression of GFP by 3-fold. This was confirmed using two different pET vectors and three E. coli strains. The authors also looked at the Translation Initiation Region (TIR), suggesting that it was generated by fusion of genetic modules without the consideration of co-evolution with E. coli ribosomes. By altering this sequence randomly, they have generated TIRs that improve production of GFP. By combining the altered sequences for the T7 promoter and the TIRs, they found plasmids capable of increasing expression of GFP and a human cancer target, MTH1. The authors mention that these alterations to the pET vectors are applicable to 88 out of the 103 versions available.

This work is very interesting for the protein research community and could benefit many labs to improve production of their proteins in the E.coli expression system. The idea of changing regions in the plasmid to improve transcription and translation could also be applied to other vectors and protein expression systems.

Although the authors have shown that the method works for both a common control protein (GFP) and also a human protein (MTH1), the affect on a wider variety of proteins would be desirable.

Author: First of all, we are grateful to the reviewer for providing feedback on our manuscript. We would like to report back that we have now tested the improved designs on another protein, the human DNA glycosylase Neil3. We have also addressed the minor points raised below.

In supplementary Figure 1, where the plasmids do not contain an asterix, does this mean that there is not a truncated T7 promoter and if so, have any of these plasmids been tested during the study?

Author: We have included a statement to clarify this situation. It reads “This design flaw is present in all pET expression plasmids containing T7lac (i.e. 88 of the 103 plasmids; Supplementary Fig. 1). In the remaining pET plasmids, the lac operator was not fused and the consensus T7 promoter (T7p^{CONS}) is intact.”

Author: We have not tested the plasmids without the lac operator in this particular study, as they do not possess the design flaw.

Some minor corrections:

Online Methods:

sfGFP fluorescence assays:

line 5: incubated at 37

line 8: medium

Generation of TIR libraries:

line 12: medium

Screening of TIR libraries:

line 5: space between 37 degrees

line 14: plasmid

SDS-PAGE and Western blotting:

line 8: transferred

Author: These minor typographical errors have been corrected.

REVIEWERS' COMMENTS:

Reviewer #1 (Remarks to the Author):

The authors have fully addressed my concerns. I appreciate the new experimental data, especially the testing of a third recombinant protein and the distribution of the recombinant proteins in the soluble and insoluble fractions. The new and improved versions of the Figures are clearer and help to better grasp the message of the work.

I believe the manuscript is now ready for publication. Congratulations to everyone involved.

Best regards,

Germán Rosano

Reviewer #2 (Remarks to the Author):

The authors have addressed most of the comments from my previous review and included one additional relevant protein, Neil3, which also benefitted from increased expression by the engineered pET vector, though as expected it did not help its solubility. I would still have liked to see a larger set of proteins tested using the improved vector but the overall paper is of interest to the protein expression community, particularly as the pET vectors are still used worldwide and will continue to be workhorses for years to come. The general idea of engineering promoter and other regions within a vector sequence to improve expression/productivity is useful information, therefore I would recommend that this paper is approved for publication.